# COVID-19 in Brazilian cities: Impact of social determinants, coverage and quality of primary health care

**Marcello Barbosa Otoni Gonçalves Guedes**[1], **Sanderson José Costa de Assis**[2], **Geronimo José Bouzas Sanchis**[2], **Diego Neves Araujo**[3], **Angelo Giuseppe Roncalli Da Costa Oliveira**[2], **Johnnatas Mikael Lopes**[4]*

1 Physical Therapy Department, Universidade Federal do Rio Grande do Norte, Rio Grande do Norte, Brazil,
2 Public Health Program, Universidade Federal do Rio Grande do Norte, Rio Grande do Norte, Brazil,
3 Faculty of Health Sciences, University Center Unifacisa, Paraíba, Brazil, 4 Medicine Department, Universidade Federal do Vale do São Francisco, Bahia, Brazil

* johnnatas.lopes@univasf.edu.br

**Data Availability Statement:** All 3 covid-19 files are available from the DATASUS database (https://covid.saude.gov.br/).

## Abstract

### Background

Brazil, as many other countries, have been heavily affected by COVID-19. This study aimed to analyze the impact of Primary health care and the family health strategy (FHS) coverage, the scores of the National Program for Improving Primary Care Access and Quality (PMAQ), and socioeconomic and social indicators in the number of COVID-19 cases in Brazilian largest cities.

### Methods

This is an ecological study, carried out through the analysis of secondary data on the population of all Brazilian main cities, based on the analysis of a 26-week epidemiological epidemic week series by COVID-19. Statistical analysis was performed using Generalized Linear Models with an Autoregressive work correlation matrix.

### Results

It was shown that greater PHC coverage and greater FHS coverage together with an above average PMAQ score are associated with slower dissemination and lower burden of COVID-19.

### Conclusion

It is evident that cities with less social inequality and restrictions of social protection combined with social development have a milder pandemic scenario. It is necessary to act quickly on these conditions for COVID-19 dissemination by timely actions with high capillarity. Expanding access to PHC and social support strategies for the vulnerable are essential.

**Funding:** The authors received no specific funding for this work.

**Competing interests:** The authors have declared that no competing interests exist.

## Introduction

The world has been suffering from the persistent evolution of SARS-COV-2 contagion that leads to the manifestation of the new coronavirus disease (COVID-19), with unprecedented consequences for public health [1]. Since January 2020, WHO has given visibility to this globalized public health problem with the Declaration on Public Health Emergency of International Concern [2]. Although Brazil has had extra time to prepare, the pandemic has hit the country and has been plaguing its population with several cases and deaths in a continuous and aggressive manner [3]. COVID-19 infection presents itself, with a significant frequency of its cases, with complicated symptoms and may lead to death [4].

Brazil is part of a large group of countries in the developing world, with a considerable portion of its population in a situation of social vulnerability [5]. The Brazilian health system (Sistema Único de Saúde—SUS), is based on the principles of universal care, comprehensive care and equity of actions based on the precept of social justice. The resources for coping with COVID-19 are structured based on these pillars [6].

SUS is essentially built by primary health care (PHC), as the first level of community care, regulating and organizing other levels and services, capable of serving most people, based on the Family Health Strategy (FHS) [7]. However, a considerable percentage of the Brazilian territory does not have PHC coverage or has a deficient quality of these services [8]. In addition, an important layer of this population is under unfavorable socioeconomic conditions, with low access to social protection policies, demonstrating a high degree of inequity in the country [5].

The quality of the services offered by PHC may directly influence the resolution of COVID-19 cases. In Brazil, there is the National Program for Improving Primary Care Access and Quality (PMAQ), which measures the level of PHC quality and promotes the quality of its services with additional resources. This program serves as an indicator for PHC to the Ministry of Health and it is based on the idea that the conditions of these services are the gateway for its users, with accountability for care, the ability to provide comprehensive care and to organize the network of health services with multidisciplinary work, promoting the highest possible degree of resolution [9].

In the Brazilian COVID-19 epidemic scenario, PHC is responsible for welcoming measures, clinical screening, social support, monitoring and care in home isolation until the discharge of social restriction, as well as clinical stabilization, effective referral and transportation to referral centers or hospital services for the most serious cases [6]. A well-qualified and engaged FHS team, for example, could provide evident support to all other health services, easing the burden on more complex levels of care [10].

In this context, in which a large part of the Brazilian population is under an unfavorable socioeconomic situation and still depends exclusively on a public system that is unable to serve the entire population with quality, the epidemiological analysis of health and social protection indicators [11], as well as the coverage and quality of PHC, become essential for the assertive coping with this epidemic [12]. The implementation of effective public policies throughout the national territory, with well-defined strategies, may improve the distribution and use of the scarce available resources.

Considering all issues presented here, the effects of this epidemic can be shown to be even more devastating in countries with great social inequities and which also make wrong choices of public policies and management strategies to combat COVID-19. Therefore, the objectives of this study were to identify the socioeconomic and health system factors associated with the accumulation of cases of COVID-19 diagnosed in the capitals of the twenty-seven Brazilian federative units (FU), characterizing them in terms of disease burden and evolutionary pattern, providing information for planning public policies for Brazil and countries with a similar profile.

## Method

This is an ecological time series study, with quantitative approach, carried out through the analysis of population-based secondary data. The outcome was COVID-19 cases confirmed by laboratory test, clinical, clinical-epidemiological and clinical-image per 100,000 inhabitants, in all Brazilian FU captaincies until the 26th epidemiological week. Outcome data were extracted from Painel Covid-19 (https://covid.saude.gov.br/) database, fed by the Health Surveillance Department and made available by the the SUS Informatics Department in [1]. To date, only criteria for confirmation of cases (laboratory and clinical-epidemiological) were used as a form of diagnosis of Covid-19, using immunological tests, rapid test or classical serology for the detection of antibodies.

Brazil is a federation with 26 states and a federal district. Each of the FU has an administrative capital that is characterized as the city with the greatest economic power in the FU. The capitals present different variability in socio-economic and demographic indicators, which helps us to understand their influences on the dispersion of COVID-19 in the population (Table 1) and described below.

**Table 1. Description of the socioeconomic characteristics of the capitals of Brazilian federative units in 2020.**

| Capitals | GDP | DD | Gini | HDI | RSP | %PHC | %FHS | N° COVID-19 | COVID-19 Incidence |
|---|---|---|---|---|---|---|---|---|---|
| Aracaju | 25185,55 | 3140,65 | 0,57 | 0,770 | 3,40 | 75,54 | 70,36 | 31777 | 1746,69 |
| Belém | 21.191,47 | 1315,26 | 0,61 | 0,746 | 4,70 | 40,79 | 22,65 | 29115 | 1177,76 |
| Belo Horizonte | 36.759,66 | 7167,00 | 0,55 | 0,810 | 1,40 | 100,00 | 80,62 | 28462 | 176,27 |
| Boa Vista | 26.752,67 | 49,99 | 0,56 | 0,752 | 6,50 | 76,84 | 50,99 | 29570 | 1501,45 |
| Campo Grande | 32.942,46 | 97,22 | 0,51 | 0,784 | 1,60 | 70,15 | 56,99 | 15953 | 135,27 |
| Cuiabá | 39.043,32 | 157,66 | 0,53 | 0,785 | 2,80 | 56,29 | 44,49 | 15345 | 429,19 |
| Curitiba | 45.458,29 | 4027,04 | 0,53 | 0,823 | 1,30 | 54,26 | 32,84 | 24170 | 137,91 |
| Florianópolis | 42.719,16 | 623,68 | 0,48 | 0,847 | ,10 | 100,00 | 88,84 | 4999 | 223,96 |
| Fortaleza | 23436,66 | 7786,44 | 0,57 | 0,754 | 3,20 | 61,36 | 49,89 | 44971 | 1238,96 |
| Goiânia | 33.004,01 | 1776,74 | 0,48 | 0,799 | 1,60 | 54,78 | 43,24 | 25853 | 338,37 |
| João Pessoa | 24319,82 | 3421,28 | 0,59 | 0,763 | 2,30 | 96,34 | 86,57 | 24597 | 1307,89 |
| Macapá | 22.181,72 | 512.902 | 0,45 | 0,733 | 9,00 | 82,17 | 53,46 | 16229 | 2368,24 |
| Maceió | 22.126,34 | 1854,10 | 0,54 | 0,721 | 6,00 | 44,60 | 26,95 | 24909 | 1273,67 |
| Manaus | 36.445,75 | 158,06 | 0,52 | 0,737 | 6,80 | 56,60 | 40,78 | 39563 | 1152,35 |
| Natal | 26497,08 | 4805,24 | 0,53 | 0,763 | 4,50 | 54,93 | 37,42 | 22041 | 863,91 |
| Palmas | 32.293,89 | 102,90 | 0,56 | 0,788 | 4,50 | 100,00 | 95,73 | 9173 | 441,28 |
| Porto Alegre | 52.149,66 | 2837,53 | 0,55 | 0,805 | 1,30 | 76,56 | 55,58 | 10753 | 114,17 |
| Porto Velho | 32.042,66 | 12,57 | 0,51 | 0,736 | 5,10 | 60,75 | 49,51 | 25006 | 1871,04 |
| Recife | 31743,72 | 7039,64 | 0,61 | 0,772 | 4,30 | 64,87 | 56,39 | 29718 | 1195,95 |
| Rio Branco | 22.287,70 | 38,03 | 0,52 | 0,727 | 4,40 | 75,63 | 49,97 | 9662 | 1485,81 |
| Rio de Janeiro | 54.426,08 | 5.265,82 | 0,48 | 0,799 | 2,60 | 47,29 | 40,96 | 80899 | 757,89 |
| Salvador | 21231,48 | 3859,44 | 0,55 | 0,759 | 3,40 | 47,53 | 36,39 | 68099 | 823,82 |
| São Luís | 27226,41 | 1215,69 | 0,53 | 0,768 | 7,00 | 45,43 | 37,26 | 17658 | 1140,50 |
| São Paulo | 58.691,90 | 7398,26 | 0,58 | 0,805 | 2,10 | 66,53 | 40,84 | 236163 | 879,29 |
| Teresina | 22481,67 | 584,94 | 0,51 | 0,751 | 3,50 | 100,00 | 100,00 | 21803 | 677,12 |
| Vitória | 73.632,55 | 3338,30 | 0,57 | 0,845 | 1,20 | 100,00 | 78,94 | 12897 | 1657,84 |

GDP: per capto gross domestic product; DD: demographic density; Gini: Gini coefficient; HDI: Human Development Index; RSP: restriction on social protection; RE: restriction on education; RH: restriction on housing; RS: restriction on sanitation; %PHC: proportion of the population covered by primary health care; %FHS: proportion of the population covered by Family Health Strategy.

In this study, time in epidemiological weeks, indicators of coverage and quality of PHC and FHS, demographic and socio-economic indicators were admitted as independent variables. PHC and FHS coverage were collected in DATASUS (https://sisaps.saude.gov.br/painelsaps/). These variables correspond to the percentage of the population covered by primary care; in this study, they were categorized as less than 50%, from 50 to 74% and 75% or higher, as recommended by the World Health Organization [5, 13].

PMAQ is a program with the objective of stimulating the evaluation process in PH C in Brazil, developed by the federal government and carried out by federal educational institutions in the country [7]. The PMAQ consists of an assessment of the conditions of infrastructure, materials, supplies and medicines in primary care services, as well as the team work process, user satisfaction and care organization, which reveal the patterns of access and quality of care. The program is divided into cycles and, in this study, data were extracted from the third cycle of the PMAQ in 2019, which cover information regarding family health teams. Teams are classified as unsatisfactory, poor, regular, good, very good, excellent [9]. However, in this study a recategorization was performed in 3 categories: low quality—with categories unsatisfactory and poor put together; medium quality—regular and good put together; and high quality—with very good and excellent teams. For access to data and more details on the calculation of this composite indicator, see: http://aps.saude.gov.br/ape/pmaq/ciclo3/.

Demographic density (DD) was collected to control bias related to the probability of human contacts in the public environment. Socioeconomic conditions were measured using per capto gross domestic product (GDP), the Human Development Index (HDI) and Gini coefficient. HDI was extracted from UNDP and Gini from PNAD 2018. HDI is an important tool to assess the development of certain locations and it is measured by the geometric mean of the sum of life expectancy at birth, education index and income index. The Gini coefficient was used as an instrument to measure the degree of income concentration in the cities, as a measure of social inequality, ranging from 0 to 1, and the closer to 1, the greater the inequality in that location. The calculation of this coefficient is given by the ratio of the areas of the Lorenz curve diagram to the accumulated income of the population. These data were collected in the National Household Sample Survey [5].

In addition to these social indicators, data was also collected on the restriction on social protection (RSP). The indicator estimate the proportion of people in situations of vulnerability. It was considered as a restriction: people who simultaneously meet the following two conditions: residents in households where there was no resident aged 14 or older who contributed to the social security institute or retired/pensioner; households with real effective household income per capita of less than ½ minimum wage, and with no members receiving income from other sources, which includes social programs. Minimum wage used as reference: R\$ 954.00 per month, in this study, ratio was calculated and stratified less than 2.60% or higher than 2.60% [5] These data were linked to the database of the Brazilian Institute of Geography and Statistics (IBGE), whose data were compiled by the Brazilian agency of the United Nations Development Program (UNDP) [5].

Statistical analysis was performed using Generalized Linear Models with an autoregressive work correlation matrix, in which the log ligand function with Gamma distribution was used. Association tests, such as Wald's chi-square test, were performed between the outcome variable and the independent variables, selecting those with "p" values equal to or less than 0.20 to be included in the adjusted model. A significance level of 5% ($\alpha < 0.05$) was adopted. The interaction variable between FHS coverage and PMAQ score was included in the analysis. In the adjusted model, the analysis was stratified by the PHC coverage strata. Data were processed using the statistical program SPSS®, version 22.0

## Results

A total of 857,741 cases were recorded in the Brazilian cities in the analyzed period, with an average of 15,884.09 (±21,604.43), revealing a heterogeneity in the dissemination of cases (Fig 1).

When observing the social and health system characteristics of the Brazilian cities, it is identified that the cities in strata of the FHS coverage, HDI and Gini presented discrepancies regarding the incidence of diagnosed cases of COVID-19, being homogeneous in the strata of PHC coverage. Cities with FHS coverage above 75% showed a lower incidence rate as well as those with HDI above 0.800 and Gini less than 0.50 (Fig 2).

When segmenting the PMAQ assessment strata according to PHC coverage, it is observed that in situations that PHC coverage is less than 50%, there are no services with a score higher than 2.99 and a similar distribution of the incidence of COVID-19. In cities with PHC coverage between 50–74%, there is a gradient of COVID-19 cases, where those with an average PMAQ score above 3 points show less load and slope. Similarly, it occurs in cities with PHC coverage above 75%, which do not have PMAQ scores below 2 points (Fig 3).

In order to estimate the effect of the independent variables and to control the effect of their interactions, an adjusted model presented in Table 2 was created, stratified by PHC coverage. In situations of PHC coverage below 50%, PMAQ score low demonstrated nine times more cases per 100,000 inhabitants COVID-19 cases than those with a score medium or high (B = 9.08; p<0.001). Positive associations were also found for RSP (B = 2.45; p<0.001), Gini (B = 7.54; p<0.001) and HDI (B = 16.19; p<0.001) with the outcome incidence rate. For the interpretation of Gini and HDI, it is more appropriate to observe the interaction of these factors and not the main effect. The Gini-HDI interaction shows a negative relationship (B = -72.46; p<0.001), which may suggest a mitigating effect in cities with high HDI and low Gini.

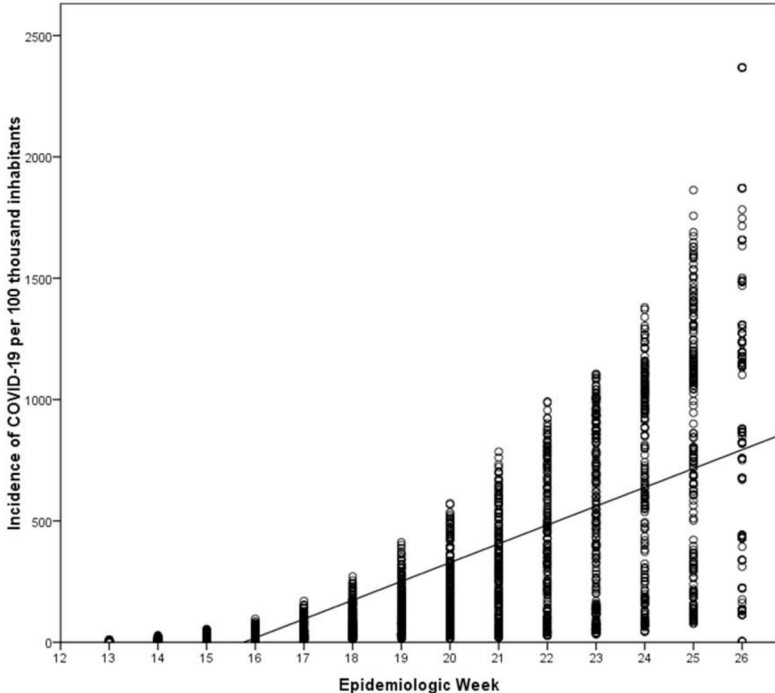

**Fig 1. Distribution of the incidence of COVID-19 cases in Brazilian cities up to the 26th epidemiological week.**
Circles are the observed cases and line is the general trend of progression of the cases.

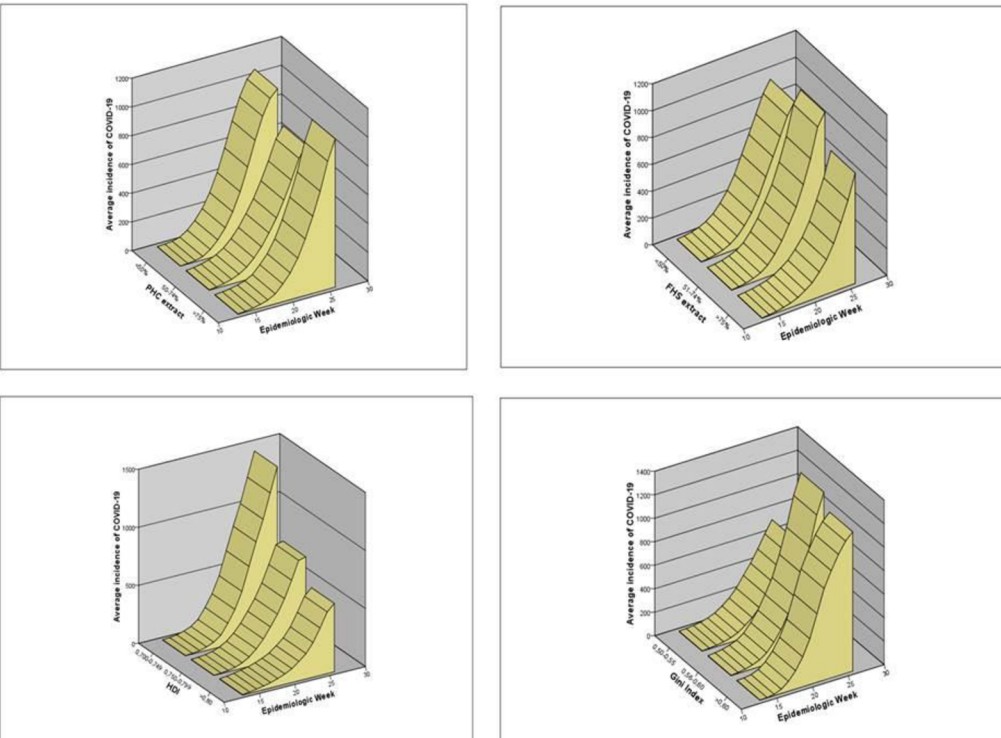

**Fig 2. Evolution of the incidence of COVID-19 cases per 100 thousand inhabitants in Brazilian cities until the 26th epidemiological week.** A-Stratified incidence by the coverage of Primary Health Care (PHC); B-Incidence stratified by the coverage of FHS; C-Stratified incidence by the HDI; D-Stratified incidence by the Gini Index.

In the PHC 50–74% stratum, cities with intermediate FHS coverage and PMAQ scores high showed almost twice less (B = 2,36) COVID-19 incidence rates than cities with lower ratings. It is still possible to verify a positive association of RSP (B = 0.33; p <0.001) with the outcome. In this PHC context, the Gini-HDI interaction shows a negative association with the outcome (B = -13.89; p<0.001), similar to cities with PHC coverage below 50% (Table 2), In other words, with each increase in the Gini-HDI interaction unit, there is a decline of more than 13 covid-19 cases per 100,000 inhabitants in the average growth rate (B = -13,89).

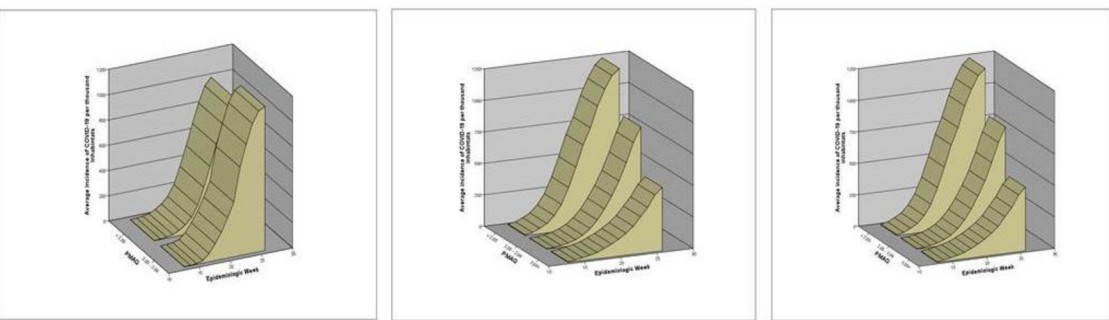

**Fig 3.** Evolution of the incidence of COVID-19 cases per 100 thousand inhabitants in Brazilian cities until the 26th epidemiological week stratified by PMAQ score when the PHC coverage is less than 50% (A), 50–70% PHC coverage (B) and>70% PHC coverage (C).

**Table 2. Adjusted model for association with the incidence of Covid-19 in the largest cities of all federal units in Brazil stratified by the coverage of primary health care.**

| Independent Variables | $B_{adj}$ | Standard Error | CI 95% Wald | | Hypothesis test | | |
|---|---|---|---|---|---|---|---|
| | | | Lower | Upper | $x^2$ Wald | df | p |
| **PHC <50% Model** | | | | | | | |
| Interception | 17.38 | 0.197 | 16.99 | 17.77 | 7786.51 | 1 | **<0.001** |
| RSP | 2.45 | 0.021 | 2.41 | 2.49 | 13543.65 | 1 | **<0.001** |
| Gini | 7.54 | 0.02 | 7.49 | 7.50 | 70924.88 | 1 | **<0.001** |
| HDI | 16.19 | 0.11 | 15.96 | 16.42 | 18704.31 | 1 | **<0.001** |
| Gini-HDI | -72.46 | 0.713 | -73.85 | -71.06 | 10304.16 | 1 | **<0.001** |
| Epidemiológical Week | 0.05 | 0.002 | 0.04 | 0.05 | 361.80 | 1 | **<0.001** |
| Stratum FHS-PMAQ | | | | | | | |
| Cov. FHS $_{<50\%}$- PMAQ$_{<low}$ | 9.08 | 0.082 | 8.92 | 9.24 | 12067.74 | 1 | **<0.001** |
| Cov. FHS $_{<50\%}$- PMAQ$_{<medium}$ | 0 | | | | | | |
| Scale | 0.62 | | | | | | |
| **PHC 50–74% Model** | | | | | | | |
| Interception | 7.06 | 1.77 | 3.58 | 10.55 | 15.81 | 1 | **<0.001** |
| RSP | 0.33 | 0.05 | 0.23 | 0.44 | 39.88 | 1 | **<0.001** |
| Gini | -8.47 | 3.36 | -15.07 | -1.88 | 6.34 | 1 | **0.01** |
| HDI | -15.49 | 6.77 | -28.76 | -2.22 | 5.23 | 1 | **0.02** |
| Gini-HDI | -13.89 | 4.38 | -22.47 | -5.30 | 10.05 | 1 | **0.002** |
| Epidemiológical Week | 0.07 | 0.004 | 0.06 | 0.08 | 306.83 | 1 | **<0.001** |
| Stratum FHS-PMAQ | | | | | | | |
| Cov. FHS $_{<50\%}$*PMAQ$_{low}$ | 2.36 | 0.17 | 2.01 | 2.71 | 177.60 | 1 | **<0.001** |
| Cov. FHS $_{<50\%}$*PMAQ$_{medium}$ | 0.85 | 0.11 | 0.62 | 1.08 | 52.29 | 1 | **<0.001** |
| Cov. FHS $_{<50\%}$*PMAQ$_{high}$ | 2.87 | 0.31 | 2.25 | 3.49 | 81.63 | 1 | **<0.001** |
| Cov. FHS$_{50-75\%}$*PMAQ$_{low}$ | 2.63 | 0.36 | 1.92 | 3.35 | 52.43 | 1 | **<0.001** |
| Cov. FHS$_{50-75\%}$*PMAQ$_{high}$ | 0 | | | | | | |
| Scale | 0.56 | | | | | | |
| **PHC> 75% Model** | | | | | | | |
| Interception | -3.15 | 1.98 | -7.05 | 0.74 | 2.51 | 1 | 0.11 |
| RSP | 0.26 | 0.07 | 0.12 | 0.41 | 12.63 | 1 | **<0.001** |
| Gini | 17.06 | 5.44 | 6.40 | 27.73 | 9.83 | 1 | **0.002** |
| HDI | -2.41 | 4.89 | -12.00 | 7.18 | 0.24 | 1 | 0.62 |
| Gini * HDI | 15.20 | 4.76 | 5.86 | 24.53 | 10.17 | 1 | **0.001** |
| Week | 0.06 | 0.003 | 0.06 | 0.07 | 442.16 | 1 | **<0.001** |
| Stratum FHS-PMAQ | | | | | | | |
| Cov. FHS $_{<50\%}$ * PMAQ$_{medium}$ | 0.76 | 0.35 | 0.07 | 1.45 | 4.71 | 1 | **0.03** |
| Cov. FHS$_{50-75\%}$ * PMAQ$_{medium}$ | -0.22 | 0.53 | -1.26 | 0.81 | 0.17 | 1 | 0.67 |
| Cov. FHS$_{>75\%}$ * PMAQ$_{medium}$ | -0.22 | 0.51 | -1.24 | 0.79 | 0.19 | 1 | 0.66 |
| Cov. FHS$_{>75\%}$ * PMAQ$_{high}$ | 0 | | | | | | |
| Scale | 0.83 | | | | | | |

B–Regression Coefficient; CI–Confidence Interval; $x^2$ –Chi-square; df–degrees of freedom; p–probability; PHC–Primary Health Care; RSP–Restriction to Social Protection; HDI–Human Development Index; Cov. FHS–Health Strategy Family Coverage; PMAQ–National Program for Improving Primary Care Access and Quality.

In cities with PHC above 75%, it was estimated that only cities with FHS coverage below 50% and PMAQ score medium have sligthly higher incidence of COVID-19 than those with higher strata of FHS and PMAQ scores (B = 0,76). Positive associations were also found for

RSP (B = 0.26; p<0.001), Gini (B = 17.06; p<0.001) and Gini-HDI interaction (B = 15.20; p<0.001). However, the HDI establishes a negative relationship with the outcome (B = -2.41) (Table 2).

In both scenarios of primary health care coverage, social indicators in the adjusted model, with emphasis on the Gini and the HDI, which present the highest coefficients in the model. On the other hand, the pure weekly evolution, without the effects of other factors, has a low influence on the evolution of cases (B<0.10). This reflects the effect of contextual factors on the spread of covid-19 in the sample.

## Discussion

This study sought to analyze the impact of PHC and FHS coverage, PMAQ scores obtained by health units, socioeconomic indicators, and social restrictions on the incidence of COVID-19 in Brazilian largest cities. We highlight that strata of better FHS coverage and PMAQ scores obtained correlation with slower progression and lower load of COVID-19, controlling the effects of socioeconomic indicators. This pandemic is also influenced by social indicators, revealing that inequality and social restrictions modulate the dispersion of cases. These factors must be considered by health managers to develop better strategies to combat the pandemic and to improve management of health resources.

Our findings pointed out that, by stratifying FHS coverage, cities with the highest coverage had their population less affected by COVID-19 cases, especially those that were better evaluated in PMAQ. These data reinforce the hypothesis that the prevention actions developed in areas better assisted by PHC services is a strategy of extreme relevance to minimize the impact of this pandemic, thus being able to assist the other levels and services of health care [14].

Adequate coverage leads to greater access to essential services, being an essential factor for a favorable scenario of health conditions within a community, either due to its high level of capillarity, or by encouraging actions of promotion and prevention for the individuals in this context [15]. These factors may directly impact prevalence and incidence of cases both in chronic non-communicable diseases and in communicable diseases, as it is the case with COVID-19 [16]. It is noteworthy that, although SUS is based on universality, not all Brazilian territory is served by PHC services [8].

In Brazil, the predominant PHC strategy is based on the community and centered on the family and the individual. Although other modalities of service organization in PHC are possible, 75% of public managers choose FHS as a structural model of their services in the primary care network [8, 17]. Thus, FHS presents itself as an important organizational model within PHC, becoming effective as a source of access to health promotion and prevention actions, as well as for the use of different health services, which can directly influence behavioral positive adjustments for the assisted population, as well as in the rationalization of health costs [17, 18].

PMAQ evaluation generates a score for each health team that is part of PHC and has joined the program [9]. In addition to the aspects related to coverage and organization of services, team quality indicators have important consequences on the population's health results. PMAQ stimulates the culture of evaluation within PHC services, in order to encourage good practices within the health units, providing the quality and innovation in its management and provision of its services with resources [10]. Our results indicated that strata with higher PMAQ scores in all PHC and FHS coverage scenarios had lower incidence of COVID-19 in Brazilian cities in the evaluated period.

PMAQ scores related findings indicate that adequate quality of PHC services can also positively impact the habits and behaviors of the registered population, which may, in general, attenuate the dissemination of the new coronavirus in Brazilian cities. Higher PMAQ scores

may imply reinforcement of actions with strategic characteristics in the FHS, for example, and in adverse situations such as this current pandemic, they become even more important. Comprehensive PHC quality assessment programs prove to be important promotion tools to increase the resolution capacity of these services [19].

Stratified and nested analyses were also carried out and some considerations are as follows. As PHC coverage strata rise, the difference in the incidence of COVID-19 between FHS-PMAQ nests becomes smaller. This organization is very favorable to support the control of this pandemic, as it is a situation of good supply of PHC services, facilitating user access to care and quality health education actions [20, 21].

PHC qualities can make a difference in coping with this pandemic. Knowledge of the territory, access, the bond between the user and the health team, comprehensive care, monitoring vulnerable families and monitoring suspicious and mild cases are fundamental strategies for containing the pandemic [6, 22]. As an example, primary care services is able to mitigate problems related to the precariousness of social and economic life, domestic violence, alcoholism and mental health problems, which also come from prolonged social isolation [22]. These roles developed by the FHS and other PHC services may stimulate popular health education and a culture of healthy habits and preventive actions to combat COVID-19.

Regarding socioeconomic indicators, cities with greater social inequality or less development had a higher number of diagnosed COVID-19 cases. It is known that high social inequality is accompanied by worse socio-demographic and health conditions for the most vulnerable social classes [23]. Often, populations under considerable social vulnerability do not have several resources or appropriate living conditions, such as access to piped water in their homes, an adequate number of people per household and economic stability. This scenario makes it difficult to carry out the prevention strategies for COVID-19, due to the population's lack of information, or even due to the lack of conditions needed for these actions to happen or to be effectively applied [24].

Another relevant issue for the analysis related to social inequities is that the prevalence of chronic diseases such as diabetes, hypertension and obesity, as well as the difficulty of its management, is higher in the poorest populations of the Brazilian society [25], and these cardiovascular conditions are one of the main risk factors for worse outcomes and mortality due to COVID-19. Even though our study did not control cardiovascular variables, a large part of the studied population is at high social risk and, therefore, preventive measures should have a special focus on these groups.

Important evidence from the results is the effect of RSP on the spread of COVID-19 in Brazilian cities. The RSP indicator portrays the absence of the State to protect citizens who are not inserted in the means of production, which increases their vulnerability. Thus, cities with a higher proportion of these individuals had more cases of COVID-19. However, these effects are minimized when PHC coverage is expanded. Based on these inferences, some social protection measures should be developed by the government in the current situation, such as 1) the provision of free and comprehensive tests, prioritizing communities that present a greater risk for the worsening and contagion of the disease, also the collection of sociodemographic data from individuals [26], 2) government officials, community leaders and the local media should consult and collaborate with specialists in medicine and public health to offer public health messages directed at low-income and high-risk populations [26]; 3) government officials must provide a minimum income for informal or unemployed workers to be able to achieve social isolation in an appropriate manner and in a timely manner; 4) alternative options for providing creative, flexible, and accessible health services to populations with difficult access to health services [27, 28]. These measures must be taken by Government officials to reduce health inequities, which makes PHC and its teams essential for the effectiveness of

social protection measures, as they are inserted in vulnerable communities, and have knowledge of the territory and the resident population and their risk factors.

Unfortunately, what we observe is a denial of the pandemic, reflecting the lack of interest by the Ministry of Health in proposing guidelines based on available scientific evidence. This reflected in the timely organization of PHC community services, which could be leveraged with appropriate investments as presented above until obtaining effective and effective vaccines for the population [28, 29].

Despite the findings that impact public health in Brazil, it is reasonable to mention some limitations that should be analyzed. The first is that COVID-19 is underdiagnosed in Brazil as well as elsewhere in the world, which leads to underreporting of events. However, this influences the disease burden in general and not on stratifications. A second limitation is the outdated ecological measure of the HDI, however we believe that its modification since the last results and the current date has been minimal. Important additional COVID-19 prevention strategies were not measured in this research and are relevant tools that should be considered in further studies.

Finally, the results of this investigation address the importance of a well-structured PHC with the FHS as a priority structural model in this fight against the coronavirus, especially in areas with considerable social inequity. Preventive actions and formal social support to the population can be decisive factors for a positive outcome in this battle. This pandemic brings unprecedented elements and specific characteristics that affect each region of the world in an unprecedented way, imposing a great challenge for the reorganization of health services across the planet.

Although each Brazilian larger city experiences specific contextual conditions and distinct moments in the development of the epidemic by COVID-19, our results allow us to make relevant generalizable and timeless analyses that will support the decision-making process of public managers. These findings are in line with the important attributions of PHC, especially those that promote a more efficient fight against the epidemic. Other countries with characteristics of social vulnerability and organization of health services similar to Brazil, can make use of these results as an aid for more effective referrals in public policies in this pandemic context.

## Author Contributions

**Conceptualization:** Marcello Barbosa Otoni Gonçalves Guedes, Sanderson José Costa de Assis, Geronimo José Bouzas Sanchis, Diego Neves Araujo, Angelo Giuseppe Roncalli Da Costa Oliveira, Johnnatas Mikael Lopes.

**Formal analysis:** Marcello Barbosa Otoni Gonçalves Guedes, Diego Neves Araujo.

**Investigation:** Marcello Barbosa Otoni Gonçalves Guedes, Sanderson José Costa de Assis, Geronimo José Bouzas Sanchis, Angelo Giuseppe Roncalli Da Costa Oliveira.

**Methodology:** Marcello Barbosa Otoni Gonçalves Guedes, Sanderson José Costa de Assis, Geronimo José Bouzas Sanchis, Johnnatas Mikael Lopes.

**Software:** Johnnatas Mikael Lopes.

**Supervision:** Marcello Barbosa Otoni Gonçalves Guedes, Angelo Giuseppe Roncalli Da Costa Oliveira, Johnnatas Mikael Lopes.

**Validation:** Johnnatas Mikael Lopes.

**Writing – original draft:** Marcello Barbosa Otoni Gonçalves Guedes, Geronimo José Bouzas Sanchis, Diego Neves Araujo, Johnnatas Mikael Lopes.

**Writing – review & editing:** Marcello Barbosa Otoni Gonçalves Guedes, Diego Neves Araujo.

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
