## [Decision Letter · Decision Letter 0]

6 May 2021

PONE-D-21-10779

COVID-19 IN BRAZILIAN CITIES: IMPACT OF SOCIAL DETERMINANTS, COVERAGE AND QUALITY OF PRIMARY HEALTH CARE

PLOS ONE

Dear Dr. LOPES,

Thank you for submitting your manuscript to PLOS ONE. After careful consideration, we feel that it has merit but does not fully meet PLOS ONE’s publication criteria as it currently stands. Therefore, we invite you to submit a revised version of the manuscript that addresses the points raised during the review process.

The manuscript has been assessed by five reviewers. Their comments are appended below. The reviewers have raised some of major concerns about the manuscript, and in particular they feel that important methodological issues exist that affect the technical soundness of your study, and the conclusions of the paper.

We look forward to receiving your revised manuscript.

Kind regards,

Bruno Pereira Nunes, Ph.D.

Academic Editor

PLOS ONE

Journal Requirements:

We note that you have stated that you will provide repository information for your data at acceptance. Should your manuscript be accepted for publication, we will hold it until you provide the relevant accession numbers or DOIs necessary to access your data. If you wish to make changes to your Data Availability statement, please describe these changes in your cover letter and we will update your Data Availability statement to reflect the information you provide.

No

No

Reviewers' comments:

Reviewer's Responses to Questions

**Comments to the Author**

1. Is the manuscript technically sound, and do the data support the conclusions?

Reviewer #1: No

Reviewer #2: Yes

2. Has the statistical analysis been performed appropriately and rigorously? 

Reviewer #1: I Don't Know

Reviewer #2: Yes

3. Have the authors made all data underlying the findings in their manuscript fully available?

Reviewer #1: Yes

Reviewer #2: Yes

4. Is the manuscript presented in an intelligible fashion and written in standard English?

Reviewer #1: Yes

Reviewer #2: Yes

5. Review Comments to the Author

Reviewer #1: The original idea to assess the role of PHC in COVID-19 incidence is good but the adopted methodological approach is very confuse and unclear.

Methods - please cite the original source of data: hospitalizations (SIH), severe cases (SIVEP-Gripe); DATASUS is the "SUS IT department); it is not a dataset.

Please expain how the score that assess the Primary Care Teams is calculated. What is evaluated?

What is "social Gini"? Please explain.

COVID-19 incidence rates were adjusted for age?

Please describe the cities included. Please describe the number of cases and the rates for each city.

Figures are not informative and hard to follow.

Where are the results of the univariate analysis?

Why authors have decided to include in the multiple model only variables with p<0.10? (too scrict criterium)

It is not clear to me if the authors intend to evaluate the influence of socioeconomic and PHC variables in the cumulative incidence of COVID-19 until the 26th week in those cities or if the authors wanted to evaluate the influence of socioeconomic and PHC variables in time-trends in incidence of COVID-19 until the 26th week in those cities.

Please clarify.

Reviewer #2: Introduction

The world has been suffering from a persistent contagion evolution by the new coronavirus (COVID-19 ),

Review: The new virus is SARS-COV-2 and COVID-19 is the disease caused of the virus.

Therefore, the objectives of this study were to identify the socioeconomic and health system factors associated with the diagnosis of COVID-19 in the main cities of each twenty-seven Brazilian federative units, characterizing them in terms of disease burden and evolutionary pattern, providing information for planning public policies for Brazil and countries with a similar profile.

Review: It does not seem to me that the aim of the study was to assess which socioeconomic factors were associated with the diagnosis of COVID-19. Associated factors could be access to health services for symptomatic cases and the number of tests offered.

Methodology

This is an ecological time series study, with quantitative approach, carried out through the analysis of population-based secondary data. The outcome was COVID-19 cases per 100,000 inhabitants, in all Brazilian main largest cities until the 26th epidemiological week.

Review: It was not clear the size of the selected cities.

It was not described in the methodology which COVID-19 case definition and database were used. In Brazil, notification of suspected cases of COVID-19 is mandatory in cases of flu like illness-ILI syndrome or severe acute respiratory syndrome-SARS. After notification, cases are investigated for confirmation of COVID-19 using the following criteria: laboratory, clinical, clinical-epidemiological and clinical-image. The databases are e-sus-VE (ILI) and Sivep-Gripe (SARS).

In the assessment of the Primary Care Teams Certification Score, teams are classified as unsatisfactory - those who have not fulfilled the commitments assumed in the adhesion;

Review. It is necessary to describe the items that were used in the assessment of the PMAQ, the Methodological Teams of Primary Care Certification Score.

In addition to these social indicators, data was also collected on the restriction on education (RE), restriction on housing (RH), restriction on sanitation (RS) and restriction on social protection (RSP).

Review: It would be important to describe which database was used to build the RH, RS and RSP indicator.

It could standardize the incidence coefficients of COVID-19, considering the different age structures in cities, we know that COVID-19 is more severe in the elderly population, with an increase in hospitalizations and deaths.

Results

A total of 857,741 cases were recorded in the Brazilian cities in the analyzed period, with an average of 15,884.09 (±21,604.43), revealing a heterogeneity in the dissemination of cases, seen in figure 1.

Review: Are the cases confirmed or suspected of COVID-19 in Brazil? SARS, flu-like illness both? What is the study period?

Figure 1. Distribution of the incidence of COVID-19 cases in Brazilian cities up to the 26th epidemiological week. Circles are the observed cases and line is the general trend of progression of the cases

Review:I didn't find figure 1.

In order to estimate the effect of the independent variables and to control the effect of their interactions, an adjusted model presented in Table 1

Review: Describe better results of the model, and how to interpret the results of beta in the incidence of the disease, when the values are greater than 1 or less. It was not clear the description of these results.

Discussion:

Review:It would be interesting to include in the discussion articles that corroborate the results. There is talk in Brazil that the Ministry of Health and governments in general

they have not trained and extensively prepared the single health system, especially primary care to track contacts and other policies that could impact the curve of the epidemic.

6. PLOS authors have the option to publish the peer review history of their article (what does this mean?). If published, this will include your full peer review and any attached files.

Reviewer #1: No

Reviewer #2: No

---

## [Author Response · Author response to Decision Letter 0]

28 Jun 2021

Firstly, we would like to thank the reviewers for their suggestions that would greatly improve the manuscript.

The document was extensively revised and readjusted on several points, as listed in the comments associated with the study specifically below.

Review Comments to the Author

Reviewer #1: The original idea to assess the role of PHC in COVID-19 incidence is good but the adopted methodological approach is very confuse and unclear.

“1. Methods - please cite the original source of data: hospitalizations (SIH), severe cases (SIVEP-Gripe); DATASUS is the "SUS IT department); it is not a dataset.”

Answer: Text was adequate as suggested. 

“Please expain how the score that assess the Primary Care Teams is calculated. What is evaluated?”

Answer: Teams are classified as unsatisfactory, poor, regular, good, very good, excellent(9). However, in this study a recategorization was performed in 3 categories: low quality - with categories unsatisfactory and poor put together; medium quality - regular and good put together; and high quality - with very good and excellent teams. The database was access in PMAQ site (http://aps.saude.gov.br/ape/pmaq/ciclo3/).

Text was adequate as suggested.

“2. What is "social Gini"? Please explain.”

Answer: The Gini coefficient was used as an instrument to measure the degree of income concentration in the cities, as a measure of social inequality, ranging from 0 to 1, and the closer to 1, the greater the inequality in that location. The calculation of this coefficient is given by the ratio of the areas of the Lorenz curve diagram to the accumulated income of the population. Theses data were collected in the National Household Sample Survey. 

Text was adequate as suggested.

“3. COVID-19 incidence rates were adjusted for age?”

Answer: We agree that age groups have an influence on the occurrence of cases sensitive to the need for health services. However, for the purposes of this study, standardization by age has no implication as the age structures of Brazilian capitals are similar. For example, in all capitals the proportion of people aged 60-64 varies from 1.3% to 2.0% and this pattern will be repeated in other age groups https://censo2010.ibge.gov.br/sinopse/index.php?uf=42&dados=26). However, it is to state that the age pattern of contagion is similar among the 27 capitals analyzed according to age.

Furthermore, our objective is to identify the effects of ecological, social, economic and health system conditions and not to compare individual conditions that might be affected by a possible distribution discrepancy between age groups.

“4. Please describe the cities included. 

Answer: The description of the capitals of the federative units (states) of brazil were placed in a tableua in the method: non-changing social characteristics in the time series and variables that changed over time.

“5. Please describe the number of cases and the rates for each city.”

Answer: Text was adequate as suggested.

“6. Figures are not informative and hard to follow.”

Answer: Text was adequate as suggested. The four graphics in Figure 1 were submitted separately to facilitate the editorial process. They are arranged from A-D.

“7. Where are the results of the univariate analysis?”

Answer: They were included in tableau 1 as they are not results directly produced in our research. This information exists in the Brazilian information system. Furthermore, our objective is to evidence the effects of the independent variables described in our method.

“8. Why authors have decided to include in the multiple model only variables with p<0.10? (too scrict criterium).”

Answer: Sorry, it was a typo. In fact, the p-value criterion to enter the model was <0.20. Text was adequate as suggested.

“9. It is not clear to me if the authors intend to evaluate the influence of socioeconomic and PHC variables in the cumulative incidence of COVID-19 until the 26th week in those cities or if the authors wanted to evaluate the influence of socioeconomic and PHC variables in time-trends in incidence of COVID-19 until the 26th week in those cities.

Please clarify.”

Answer: Data were analyzed using a matrix of autoregressive correlations in the GEE method. This approach allows us to verify the effects of independent factors on the outcome trend. As the trend of cases in the analyzed period is increasing (week factor), factors that influence the trend have implications for the increase in accumulated cases.

Reviewer #2

“Introduction

1. The world has been suffering from a persistent contagion evolution by the new coronavirus (COVID-19 ),

Review: The new virus is SARS-COV-2 and COVID-19 is the disease caused of the virus.”

Answer: Text was adequate as suggested.

“2. Therefore, the objectives of this study were to identify the socioeconomic and health system factors associated with the diagnosis of COVID-19 in the main cities of each twenty-seven Brazilian federative units, characterizing them in terms of disease burden and evolutionary pattern, providing information for planning public policies for Brazil and countries with a similar profile.

Review: It does not seem to me that the aim of the study was to assess which socioeconomic factors were associated with the diagnosis of COVID-19. Associated factors could be access to health services for symptomatic cases and the number of tests offered.”

Answer: After the analysis, we adjusted the objective. Therefore, the objectives of this study were to identify the socioeconomic and health system factors associated with the accumulation of cases of COVID-19 diagnosed in the capitals of the twenty-seven Brazilian federative units (FU).

“Methodology

1.This is an ecological time series study, with quantitative approach, carried out through the analysis of population-based secondary data. The outcome was COVID-19 cases per 100,000 inhabitants, in all Brazilian main largest cities until the 26th epidemiological week. Review: It was not clear the size of the selected cities.”

Answer: The selected cities are the capitals of the 27 federated states of Brazil. Text was adequated.

“2. It was not described in the methodology which COVID-19 case definition and database were used. In Brazil, notification of suspected cases of COVID-19 is mandatory in cases of flu like illness-ILI syndrome or severe acute respiratory syndrome-SARS. After notification, cases are investigated for confirmation of COVID-19 using the following criteria: laboratory, clinical, clinical-epidemiological and clinical-image. The databases are e-sus-VE (ILI) and Sivep-Gripe (SARS).”

Answer: Text was adequate to meet the suggestions.

“In the assessment of the Primary Care Teams Certification Score, teams are classified as unsatisfactory - those who have not fulfilled the commitments assumed in the adhesion;

Review. It is necessary to describe the items that were used in the assessment of the PMAQ, the Methodological Teams of Primary Care Certification Score.”

Answer: Text was adequate to meet the suggestions.

“In addition to these social indicators, data was also collected on the restriction on education (RE), restriction on housing (RH), restriction on sanitation (RS) and restriction on social protection (RSP).

Review: It would be important to describe which database was used to build the RH, RS and RSP indicator.”

Answer: Text was adequate to meet the suggestions.

“It could standardize the incidence coefficients of COVID-19, considering the different age structures in cities, we know that COVID-19 is more severe in the elderly population, with an increase in hospitalizations and deaths.”

Answer: We agree that age groups have an influence on the occurrence of cases sensitive to the need for health services. However, for the purposes of this study, standardization by age has no implication as the age structures of Brazilian capitals are similar. For example, in all capitals the proportion of people aged 60-64 varies from 1.3% to 2.0% and this pattern will be repeated in other age groups https://censo2010.ibge.gov.br/sinopse/index.php?uf=42&dados=26). However, it is to state that the age pattern of contagion is similar among the 27 capitals analyzed according to age.

Furthermore, our objective is to identify the effects of ecological, social, economic and health system conditions and not to compare individual conditions that might be affected by a possible distribution discrepancy between age groups.

“Results

1. A total of 857,741 cases were recorded in the Brazilian cities in the analyzed period, with an average of 15,884.09 (±21,604.43), revealing a heterogeneity in the dissemination of cases, seen in figure 1.

Review: Are the cases confirmed or suspected of COVID-19 in Brazil? SARS, flu-like illness both? What is the study period?”

Answer: These are all confirmed cases. The period refers to the 13th and 26th epidemiological weeks.

“2. Figure 1. Distribution of the incidence of COVID-19 cases in Brazilian cities up to the 26th epidemiological week. Circles are the observed cases and line is the general trend of progression of the cases

Review:I didn't find figure 1.”

Answer: It was probably an error in the submission system. It is found at the end of the manuscript.

“3. In order to estimate the effect of the independent variables and to control the effect of their interactions, an adjusted model presented in Table 1

Review: Describe better results of the model, and how to interpret the results of beta in the incidence of the disease, when the values are greater than 1 or less. It was not clear the description of these results.”

Answer: Text was adequate to meet the suggestions.

Discussion:

1. “Review:It would be interesting to include in the discussion articles that corroborate the results. There is talk in Brazil that the Ministry of Health and governments in general 

they have not trained and extensively prepared the single health system, especially primary care to track contacts and other policies that could impact the curve of the epidemic.”

Answer: Text was adequate to meet the suggestions.

---

## [Decision Letter · Decision Letter 1]

13 Aug 2021

PONE-D-21-10779R1

COVID-19 IN BRAZILIAN CITIES: IMPACT OF SOCIAL DETERMINANTS, COVERAGE AND QUALITY OF PRIMARY HEALTH CARE

PLOS ONE

Dear Dr. LOPES,

Thank you for submitting your manuscript to PLOS ONE. After careful consideration, we feel that it has merit but does not fully meet PLOS ONE’s publication criteria as it currently stands. Therefore, we invite you to submit a revised version of the manuscript that addresses the points raised during the review process.

The reviewer has raised some of minor concerns about the manuscript, and there are some methodological issues that affect the technical soundness of your study.

We look forward to receiving your revised manuscript.

Kind regards,

Bruno Pereira Nunes, Ph.D.

Academic Editor

PLOS ONE

Journal Requirements:

Additional Editor Comments (if provided):

Reviewers' comments:

Reviewer's Responses to Questions

**Comments to the Author**

1. If the authors have adequately addressed your comments raised in a previous round of review and you feel that this manuscript is now acceptable for publication, you may indicate that here to bypass the “Comments to the Author” section, enter your conflict of interest statement in the “Confidential to Editor” section, and submit your "Accept" recommendation.

Reviewer #2: (No Response)

2. Is the manuscript technically sound, and do the data support the conclusions?

Reviewer #2: Yes

3. Has the statistical analysis been performed appropriately and rigorously? 

Reviewer #2: Yes

4. Have the authors made all data underlying the findings in their manuscript fully available?

Reviewer #2: (No Response)

5. Is the manuscript presented in an intelligible fashion and written in standard English?

Reviewer #2: Yes

6. Review Comments to the Author

Reviewer #2: Methods - 1. The COVID-19 reporting systems where data were collected were not detailed. Are they mild and serious cases? Were the systems used e sus ve (mild) and sivepripe (severe)?

2. The standardization of the incidence coefficient would be important, due to the difference in the impact of the disease in the elderly. The proportion of elderly people (> = 60 years) varies from 6.9% in Boa Vista to 20.4% in Porto Alegre, according to data from DATASUS in 2020.

3. As the study was carried out until week 26 of 2020, there were only criteria for confirmation of cases (laboratory and clinical-epidemiological). (Source: file:///C:/Users/AnaRibeiro/Downloads/GuiaDeVigiEp-final.pdf). After 8/5/2020 (SE 31), the new clinical and clinical-image criteria were included (https://portalarquivos.saude.gov.br/images/af_gvs_coronavirus_6ago20_adjustments-finalis-2.pdf).

Results.

Figure 1 shows the image per city per 100,000 inhabitants per city. What is considered each circle. Which cities are shown.

table 1: I do not understand some results. In situations of PHC coverage below 50%, PMAQ score low demonstrated "nine times more cases" per 100,000 inhabitants COVID-19 cases than those with a score medium or high (B=9.08; p<0.001). In the PHC 50-74% stratum, cities with intermediate FHS coverage and PMAQ scores high showed almost "twice less" (B=2,36) COVID-19 incidence rates than cities with lower ratings.

For the interpretation of Gini and HDI, it is more appropriate to observe the interaction of these factors and not the main effect. The Gini-HDI interaction shows a negative relationship (B=-72.46; p<0.001), which may suggest a mitigating effect in cities with high HDI and low Gini.

7. PLOS authors have the option to publish the peer review history of their article (what does this mean?). If published, this will include your full peer review and any attached files.

Reviewer #2: No

---

## [Author Response · Author response to Decision Letter 1]

20 Aug 2021

EDITOR

Editor-in-Chief

Plos One

 Aug 20, 2021.

Dear Editor,

Thank you for your email with the reviewers’ comments. We have reviewed the comments and edited the manuscript accordingly. Please, find attached our point-by-point response to the reviewers. All authors have read this protocol and agreed with Plos One policy. We hope the revised manuscript is now suitable for publication.

Sincerely. Johnnatas Mikael Lopes.

Reviewer Comments:

Comments to the Author:

1. If the authors have adequately addressed your comments raised in a previous round of review and you feel that this manuscript is now acceptable for publication, you may indicate that here to bypass the “Comments to the Author” section, enter your conflict of interest statement in the “Confidential to Editor” section, and submit your "Accept" recommendation.

Reviewer #2: (No Response)

2. Is the manuscript technically sound, and do the data support the conclusions?

Reviewer #2: Yes

3. Has the statistical analysis been performed appropriately and rigorously?

Reviewer #2: Yes

4. Have the authors made all data underlying the findings in their manuscript fully available?

Reviewer #2: (No Response)

Response: All data used for this manuscript is in the public domain. In the Methods section you will find all this information, as well as links to access the full material.

5. Is the manuscript presented in an intelligible fashion and written in standard English?

Reviewer #2: Yes

6. Review Comments to the Author

Reviewer #2:

Methods –

1. The COVID-19 reporting systems where data were collected were not detailed. Are they mild and serious cases? Were the systems used e sus ve (mild) and sivepripe (severe)?

Response: Thank you for your comments. In the present study, no differentiation was made between severe and mild cases. Cases diagnosed by COVID-19 were used, according to data extracted from the Covid-19 Panel (https://covid.saude.gov.br/) database, fed by the Health Surveillance Department and made available by the SUS Informatics Department. This information can be found in the methods section of the manuscript.

2. The standardization of the incidence coefficient would be important, due to the difference in the impact of the disease in the elderly. The proportion of elderly people (> = 60 years) varies from 6.9% in Boa Vista to 20.4% in Porto Alegre, according to data from DATASUS in 2020.

Response: We agree with the reviewer that the higher proportion of elderly people in the capitals of the southern region of the country make them more likely to have COVID-19 cases. However, this difference in reported proportion does not match official data from the Brazilian Institute of Geography and Statistics, the official body for this information (https://censo2010.ibge.gov.br/sinopse/index.php?uf=43&dados=26#topo_piramide).

In any case, the objective of the research is not to estimate the differences in the load of COVID-19 between the capitals, where standardization would be of great importance, as the age composition would impact the compared estimates.

The objective was to estimate the effects of socioeconomic conditions and health system organization prior to the pandemic on the occurrence of COVID-19 cases. In this facet, the effect of age composition is reduced in the inferences, as the comparison is made with levels of independent variables, such as coverage of the PHC and FHS, which do not suffer a direct effect from the age composition. Furthermore, they assume that the primary health care system must be organized according to the local population profile.

Thus, the inferential analysis does not compare the Brazilian capitals, but analyzes them as a single group, being stratified by factors such as healthcare coverage. 

3. Como o estudo foi realizado até a semana 26 de 2020, havia apenas critérios para confirmação dos casos (laboratoriais e clínico-epidemiológicos). (Fonte: arquivo: /// C: /Users/AnaRibeiro/Downloads/GuiaDeVigiEp-final.pdf). Após 05/08/2020 (SE 31), os novos critérios clínicos e de imagem clínica foram incluídos (https://portalarquivos.saude.gov.br/images/af_gvs_coronavirus_6ago20_adjustments-finalis-2.pdf).

Response: Thank you for your comments. The following sentences were added:

To date, only criteria for confirmation of cases (laboratory and clinical-epidemiological) were used as a form of diagnosis of Covid-19, using immunological tests, rapid test or classical serology for the detection of antibodies.

Results.

4. Figure 1 shows the image per city per 100,000 inhabitants per city. What is considered each circle. Which cities are shown.

Response: Figure 1 shows a general graph of the incidence in each city analyzed (circles) in order to show the exponential evolutionary profile in all of them.

5. table 1: I do not understand some results. In situations of PHC coverage below 50%, PMAQ score low demonstrated "nine times more cases" per 100,000 inhabitants COVID-19 cases than those with a score medium or high (B=9.08; p<0.001). In the PHC 50-74% stratum, cities with intermediate FHS coverage and PMAQ scores high showed almost "twice less" (B=2,36) COVID-19 incidence rates than cities with lower ratings.

Response: The first interpretation is correct. This is equivalent to stating that in situations of low PHC coverage, having better quality of care had an impact on the dissemination of COVID-19. On the other hand, the second statement is based on the inverse interpretation of the model's coefficient relative to the worst-case coverage and quality of care. This inversion of interpretation may have led to confusion.

6. For the interpretation of Gini and HDI, it is more appropriate to observe the interaction of these factors and not the main effect. The Gini-HDI interaction shows a negative relationship (B=-72.46; p<0.001), which may suggest a mitigating effect in cities with high HDI and low Gini.

Response: In multivariate analyses, when there is an interaction between factors, their interpretation prevails over that of isolated factors, which may contain measurement biases.

7. PLOS authors have the option to publish the peer review history of their article (what does this mean?). If published, this will include your full peer review and any attached files.

Response: Yes, the authors choose to publish the peer review history of this article

All changes made are highlighted in the manuscript.

Thank you for your comment. The manuscript has been revised accordingly. 

Sincerely,

Johnnatas Mikael Lopes

---

## [Editor Report · Decision Letter 2]

31 Aug 2021

COVID-19 IN BRAZILIAN CITIES: IMPACT OF SOCIAL DETERMINANTS, COVERAGE AND QUALITY OF PRIMARY HEALTH CARE

PONE-D-21-10779R2

Dear Dr. LOPES,

We’re pleased to inform you that your manuscript has been judged scientifically suitable for publication and will be formally accepted for publication once it meets all outstanding technical requirements.

Kind regards,

Bruno Pereira Nunes, Ph.D.

Academic Editor

PLOS ONE
---

## [Editor Report · Acceptance letter]

9 Sep 2021

PONE-D-21-10779R2 

COVID-19 IN BRAZILIAN CITIES: IMPACT OF SOCIAL DETERMINANTS, COVERAGE AND QUALITY OF PRIMARY HEALTH CARE 

Dear Dr. LOPES:

I'm pleased to inform you that your manuscript has been deemed suitable for publication in PLOS ONE. Congratulations! Your manuscript is now with our production department. 

Kind regards, 

on behalf of

Dr. Bruno Pereira Nunes 

Academic Editor

PLOS ONE